# Optimization of inner panel thickness for enhanced stiffness and vibration control in car door assemblies

Pandurang Maruti Jadhav[1]*, Kishor B. Waghulde[1], Venushree Khanke[2], Rupesh V. Bhortake[3], Prasad D. Kulkarni[4], Mohammad israr[5], Muazu Jibrin Musa [6]*, Md Irfanul Haque Siddiqui[7], Subhav Singh[8,9], Deekshant Varshaney[10,11]

**1** Department of Mechanical Engineering, Dr. D. Y. Patil Institute of Technology, Pimpri, Pune, Maharashtra, India, **2** Department of Civil Engineering, AISSMS College of Engineering, Pune, Maharashtra, India, **3** Department of Mechanical Engineering, Marathwada Mitramandal's Institute of Technology, Pune, Maharashtra, India, **4** Department of Mechanical Engineering, Annasaheb Dange College of Engineering and Technology, Ashta, Maharashtra, India, **5** President, Maryam Abacha American University of Nigeria Hotoro GRA, Kano State, Federal Republic of Nigeria, **6** Department of Computer Engineering, Maryam Abacha American University of Niger, Maradi, Republic of Niger, **7** Department of Mechanical Engineering, College of Engineering, King Saud University, Riyadh, Saudi Arabia, **8** Chitkara Centre for Research and Development, Chitkara University, Himachal Pradesh, Solan, India, **9** Centre for Promotion of Research, Graphic Era (Deemed to be University), Uttarakhand, Dehradun, India, **10** Centre of Research Impact and Outcome, Chitkara University Institute of Engineering and Technology, Chitkara University, Rajpura, Punjab, India, **11** Division of Research and Development, Lovely Professional University, Punjab, Phagwara, India

* mmjibrin@maaun.edu.ne (MJM); pmjadhav@rediffmail.com (PMJ)

## Abstract

The inner panel of the side door acts as a backbone of the door assembly. It is designed to accommodate all the parts required to complete the door assembly and its intended functions. The stiffness of the inner panel indirectly protects the inside passengers from weather conditions and side impacts. The inner panel, outer panel, and a few stiffener plates comprise the door's structure. The preferred method of hemming is used to connect the inner panel and outer panel at their periphery. In hemming, the crimping of the inner panel is done by using the "U" shape of the outer panel at its periphery. The inner panel and the outer panel together will have a cutout, which will provide clear visibility to both inside passengers and outsiders through this cutout cum window. The closing and opening of this window will be controlled by the electric motor-operated window glass regulator mechanism. The assessment of the Swift Dzire side door assembly was carried out to determine the impact of the inner panel thickness variations on the dynamic characteristics of the door assembly. Effects have been studied and tried out to establish the relationship with thickness. In general, the car users or passengers and the door are in contact. So, the dynamic parameters of the door assembly have been studied, which are related to comfort. These parameters are natural frequencies, mode shapes, dynamic stiffness, and vibrations of the outer panel within the frequency band of 0 Hz to 100 Hz. Mainly, the

**Data availability statement:** All relevant data are within the paper and its Supporting Information files.

**Funding:** The author(s) received no specific funding for this work.

**Competing interests:** The authors have declared that no competing interests exist.

research study was performed on the baseline inner panel of 1.367 mm thick, and thickness variations were studied between 1.0 mm to 2.4 mm, with an increment of 0.2 mm. The door outer panel/surface dynamic stiffness and vibrations were investigated at 15 critical locations. Based on the output, dynamic results such as modal frequency, mode shape, local dynamic stiffness, and surface vibrations. A relationship has been established between these dynamic parameters and thickness variation of the inner panel, and finally, the optimal thickness has been suggested to meet the required targets.

## 1. Introduction

After the COVID-19 pandemic, cars are increasingly used in both metropolitan cities and small towns. The car can provide transportation and isolation for a small family. The car door is used to get in and out of the car. The car door assembly includes structural panels, plastic trims, hinges, latch, window glass, window glass operating mechanism and its electric motor, side mirror, controls switch, wire harness and weather seals.

The door assembly is a component of the car body. It safeguards inside passengers from extreme weather conditions, including cold, rain, hot summers, and storms. In addition to the weather conditions, it also protects passengers from side impacts and accidents. The dynamic behaviors and strength of the door primarily depend on its structural panels. The dynamic behavior of door includes its natural frequency, mode shape, dynamic stiffness, vibration, and noise. These dynamic behaviors are primarily influenced by the design and the thickness of the inner panel, which serves as the backbone of the door assembly. The aim of this research is to establish the relationship between inner panel thickness and dynamic behavioral parameters, aiming to find the optimal thickness for the specified targets (Table 1) of the side door assembly.

Research papers published between 2002 and 2022 were reviewed to understand the current status of research on the side door and its inner panel. A comprehensive analysis of 81 technical papers were summarized and published in a review paper, primarily this assessment was done to identify the work related to the door inner panel and find the research gaps [1]. Here is a summary of a few key technical papers related to door dynamics.

Hao Chen et al [2] identified the higher sensitive panels of the test door using a surface response method. FE-based simulation results of the door were verified with physical test data. The thickness variation of the door panels was used to list the higher sensitive panels to the fundamental modes. Modal frequencies were optimized based on the most sensitive surface response and thickness variations. Even after the reduction of 5.74% mass, the modal frequencies are the same.

Wentao Yu [3] analyzed interior noise issue of rough asphalt road for constant speed at 40 km/h was carried out. The back door and car cabin air cavity coupled frequency response was investigated in detail and concluded. The easy methods of simulation and experimental test were used, and the author added the dynamic

**Table 1. Dynamic Parameter Target of the Door Assembly.**

| Parameter | Target |
|---|---|
| 1st Global Lateral Bending Mode | > 26 Hz |
| 2nd Global Lateral Bending Mode | > 42 Hz |
| Min. Dynamic Stiffness between 20 Hz to 60 Hz | > 8.5 N/m |
| Max. Acceleration between 0 Hz to 100 Hz | < 3.0 m/sec$^2$ |

absorbers for de-coupling the back of the door and cab cavity modes. Due to the addition of dynamic absorbers, the coupled frequency of 43.1 Hz was reduced to 37.2 Hz, and 8 dB interior noise was reduced at 42 Hz, while vehicles were running on rough roads at a constant speed of 40 km/h.

XIQUAN QI et al [4] verified the finite element modal analysis results through an experimental test conducted within a semi-anechoic chamber. The first two global modes and their responses were correlated. Based on the transfer function, the FE model was verified, and it was observed that the transfer function poles and modal frequencies of a door were the same.

WII Wan I Mirza et al [5] carried out the scientific study on FRF point optimization of the door outer surface. Generally, more time is required to find the correct location for getting an experimental modal response for the better correlation of mode shapes. The author used the effective impedance method (EIM) for the selection of 30 points on the car door surface. EIM is an accurate and time-saving method in finite element analysis in the case of complex systems.

Chandru B.T. et al [6] studied the experimental modal test and free-free finite element (FE) modal analysis. The results have better agreements between them. It predicts that finite element simulation is close to reality. Based on validated results of the correlated FE model, the author has reduced the vibration by design modifications. Mohan Kumar G R [7] confirms the finite element model using physical experimental modal test mode shapes and corresponding the frequency values, which matched closely. Based on the validated FE modal analysis, the author suggested design modifications, which are working well to improve the modal frequencies.

Neelappagowda Jagali et al [8] studied the damper effect in comparison to test and FEA results. Sunil S. Patil et al [9] analyzed door slams under transient conditions and validated them by FEA. Test results of stress, buckling energy, strain, and acceleration compared with FEA outputs. Gao Yunkai et al [10] developed a method to correlate the door collision test results with BEM outputs and the same method was used to verify the other door designs without a test. Erkut Yalcın et al [11] studied a vibro-acoustic transient simulation method for noise calculation. The exterior noise of a door slam is compared with test in time domain. A. Lucifredi et al [12] demonstrated a good correlation between the door noise pressure and vibro-acoustic analysis and an experimental test.

D. A. Desai et al [13] analyzed interior noise of a simplified model of a car tested in an anechoic chamber and is compared with vibro-acoustic FEA. It is done for door impact load. Chunlong Ma et al [14] optimized the parameters related to the wind glass sliding to reduce the squeak and rattle noise. Hyeonho JO et al [15] developed a sound quality matrix to assess the sound quality parameters of the door latch noise. Yoshiaki ITOH et al [16] created an algorithm cum filtering method to analyze vibrations in the time domain and frequency domain. Sajjad Beigmoradi et al [17] worked on the gap between the door and its trim to control the squeak and rattle noise by using the statistical energy method.

A. Maressa et al [18] tried to improve vehicle level noise using topography optimization for panels. Niermann J et al [19] analyzed transient behaviors of vehicle noise using Wavelet and Short Time Fourier Transformation. Mihir H. Patel et al [20] carried out an analysis of variable sample data of real-time test using FFT to find fundamental frequencies. Yashas S [21] did a modal and static analysis of door trim using FE-based and calculated frequencies and stresses respectively. Chandru B. T et al [22] calculated the front door frequency of five different models and compared it with engine idle frequency.

Said Darwish et al. [23] studied the rear door's lateral stiffness and vertical displacement, which were controlled below the target with added reinforcement. M. Grujicic et al [24] did the door optimization by considering multidisciplinary loadings such as durability, NVH, manufacturing, and crashworthiness. Rui Feng et al [25] used a reanalysis method, and an algorithm was developed using initial parameters and used to analyze and optimize the new structure of the door.

LIU Xing [26] was briefed on the vehicle's NVH performance improvement based on damping treatment, body contributions, and hard point analysis.

Liu Ying-jie et al. [27] developed an interior noise calculation methodology based on the Lighthill analogy using Computational Fluid Dynamics (CFD) and the Finite Element Method (FEM). P. M. Jadhav et al [28] established the correlation of test and CAE results and isolator design for door vibration improvement.

According to the literature analysis, car door designs have been assessed over the past 20 years based on modal frequencies, vibration levels, and noise levels. The majority of the work is conducted using finite element (FE) analysis and physical experimental modal testing. There are some papers on the evaluation of door sag, side impact, and door hinges. To date, none of the authors have evaluated the inner panel thickness or its contribution to door performance. Therefore, it is decided to assess the door inner panel sensitivity.

The focus of this scientific work is to verify the contribution of the variation in inner panel thickness to door dynamic behaviors. Establish the relationship between dynamic behaviors and the thickness of the inner panel and finally optimize it. This will indirectly result in the development of a methodology to establish the relationship between door dynamic parameters and inner panel thickness, and to optimize the inner panel. This scientific work is done mainly to quantify the effects of thickness change on the dynamic behaviors of the door assembly in terms of modal frequency, dynamic stiffness, and vibrations between the critical frequency bands of 0 Hz to 100 Hz. The research was conducted on a finite element (FE) model of the door assembly (Fig 1), which was the result of previous correlation work [28].

## 2. Theoretical aspects used for evaluation

### 2.1. Modal and frequency response function

The theoretical part of modal analysis and frequency response analysis is used to investigate the required research work related to the inner panel of the door. Modal analysis is a method of calculating the frequency value and its mode shape

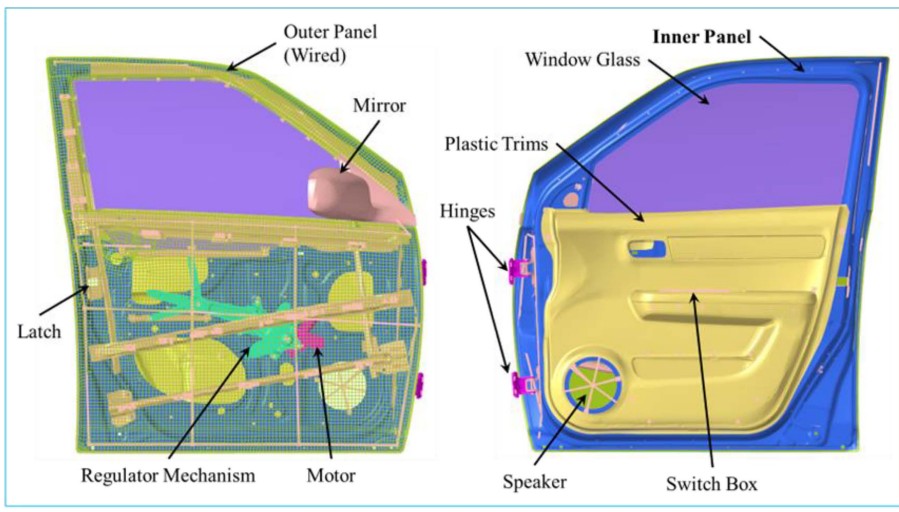

**Fig 1. Door Assembly.**

vector. Theoretically, modal analysis is similar to the wave equation, which calculates the dynamics of a vibrating string. Finally, based on the wave equation, the equation of motion is used to represent the dynamic system in mathematical form, accounting for the mass, damping, and stiffness of the system. The mathematics used to solve partial differential equations are the basis of this equation of motion [29], which covers almost all continuous dynamic structural systems or frequency response.

$$M\ddot{X}(t) + C\dot{x}(t) + Kx(t) = f(t) \qquad (1)$$

Where, $M$ is the mass matrix, $\ddot{X}(t)$ is the acceleration vector, $C$ is the damping matrix, $\dot{x}(t)$ is the velocity vector, $K$ is the stiffness matrix, $x(t)$ is the displacement or response vector, $f(t)$ is the excitation vector, and $(t)$ is the time variation.

Natural frequencies and mode shapes are independent of external loads, the system-damping effect is ignored to avoid numerical complexity, and then, under the free-free and constrained modal analyzed. The simplified equation for modal analysis will look like as one below.

$$M\ddot{X}(t) + Kx(t) = 0 \qquad (2)$$

This is the second-order equation of constant coefficient of the linear differential equation and solution of it is

$$x(t) = \varnothing e^{i\omega t} \qquad (3)$$

Where, $w_1, w_2, ...., w_n$ represent $n^{th}$ natural frequency and $\varnothing_1, \varnothing_2, ...., \varnothing_n$ represent the its corresponding mode shapes. From the above equation (2), the simplified equation of natural frequency is calculated as below.

$$Wn = \sqrt{k/m} \qquad (4)$$

Where, $m$ is system mass and $k$ system stiffness.

When a system is externally excited or impacted by a hammer, the entire system gets excited and responds to it. These responses can be measured by accelerometers anywhere on the structural system in the frequency domain. These responses are the exchange of load and structural resistance between these two points, i.e., the excitation point and corresponding response point. To get a clear understanding and find out modal parameters, a simplified system of single degree of freedom (SDOF) is used, which consists of a spring-mass-damper as shown below. The response of this simplified system, as a function of frequency (Fig 2), has a peak response $x(t)$ at its natural frequency ($Wn$) due to external force $f(t)$ excitation.

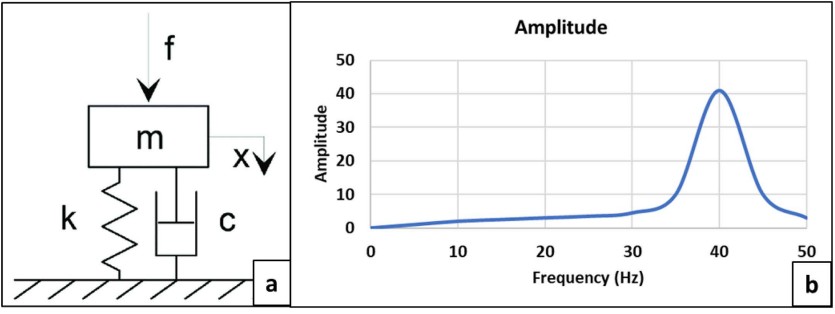

**Fig 2. (a) SDOF System and (b) Frequency Response.**

## 2.2. Trend line of linearly dependent data by least square method

Modal frequency values of global modes are considered linear because the inner panel material has linear properties. Using the global mode frequency value of the respective mode, a 2D graph of frequency vs thickness of each mode is plotted. The trend lines of each global mode are plotted/calculated using the least squares (LS) method [29], which is used to fit a mathematical function through given data. The trend line gives a theoretical understanding of the mechanism responsible for the thickness change. These linear trend lines will be expressed by the equation below.

$$Y_t = bt + a \qquad (5)$$

Where, $Y_t$ is a projected value of frequency value for a selected value of a thickness (t), $Y$ is an intercept, b is the slope of the line, and t is any selected thickness.

## 2.3. Linear interpolation or regression

The thickness range for the sensitivity study was selected within the design and manufacturing constraints. Multiple thicknesses were selected between 1.0 mm and 2.4 mm with steps of 0.2 mm, (1.0, 1.2, 1.4, 1.6, 1.8, 2.0, 2.2, and 2.4) mm, and the analysis was performed. All outcomes were compared against specified targets. Here, the results of any one thickness did not meet all targets. Therefore, we need to calculate the optimum thickness that will satisfy all requirements satisfactorily. Linear interpolation is employed [30]. By using this method, the intermediate thickness value of the inner panel is calculated using equation (6) and Fig 3 below.

$$y = y_0 + \frac{(x - x_0)(y_1 - y_0)}{(x_1 - x_0)} \qquad (6)$$

Where,
$x_0$ and $y_0$ are the first values of dynamic parameters and thickness respectively,
$x_1$ and $y_1$ are the second values of dynamic parameter and thickness respectively.
$x$ is the dynamic parameter/ target value, where the interpolation is performed.
$y$ is the thickness value obtained after interpolation.

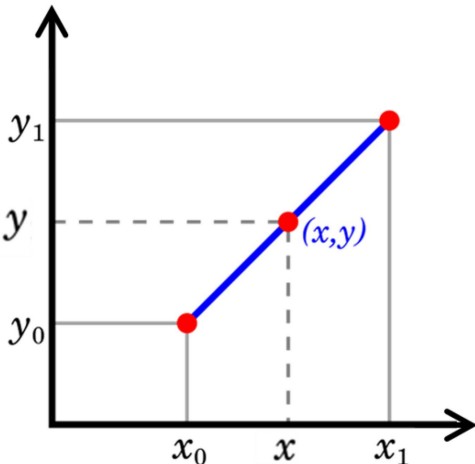

**Fig 3. Graphical Representation of Linear Interpolation.**

## 3.  Methodology of Door Inner Panel Effect Evaluation

The inner panel design that is currently in place for Maruti Swift Dzire's inner door is 1.367 mm thick (Gauge 17). To study the effect of inner panel design and thickness (Figs 1 and 4) on door assembly dynamic behaviors, the thickness was varied from 1.00 mm to 2.40 mm in steps of 0.2 mm. These thicknesses are depicted as; T10 = 1.0 mm, T12 = 1.2 mm, T1.4 = 1.367 mm, T16 = 1.6 mm, T18 = 1.8 mm, T20 = 2.0 mm, T22 = 2.2 mm, and T24 = 2.4 mm. With these thicknesses, on the door assembly, performed modal analysis to calculate natural frequencies and corresponding mode shapes, and FRF analysis to calculate local stiffness by exciting at the same location, and to calculate acceleration FRF at critical locations (Fig 4) by exciting at hinge locations.

### 3.1.  Modal Analysis Results Evaluation Method

The modal analysis of door assembly was performed under free-free conditions on all eight models, and the results, in terms of frequency and mode, were tabulated. The global mode or global bending modes of the door were listed and compared. The main focus was on tracking the first two global modes of each behavior. Global mode means the entire door assembly is geometrically involved in shaping the mode shape or deformation of any mode, which is dominantly bending in a specific direction. Generally, fundamental global bending modes are considered as torsional bending, vertical bending, and lateral bending.

In our case, lateral bending modes are included in the optimization specifications and are considered one of the parameters. The frequency targets for the first and second lateral bending mode optimization were 26 Hz and 42 Hz, respectively.

To confirm the status of all eight models' global modes, frequency values were recorded and displayed on the a 2D graph. Where the X axis has thickness and the Y axis has frequency values. A linear trend line was established for each global blending mode by utilizing this 2D curve. These linear trend lines will be utilized to determine the optimal thickness for a frequency target of 26 Hz and 42 Hz.

### 3.2.  Dynamic stiffness results evaluation method

To assess the dynamic stiffness of the door assembly, the outer surface/ panel was spatially divided into 15 critical locations/ points (Fig 4), such as D1, D2, to D15. Using the hinge excitations, all of these points were evaluated, and the frequency response function (FRF) was calculated at the same location from 0 Hz to 100 Hz. From the analysis,

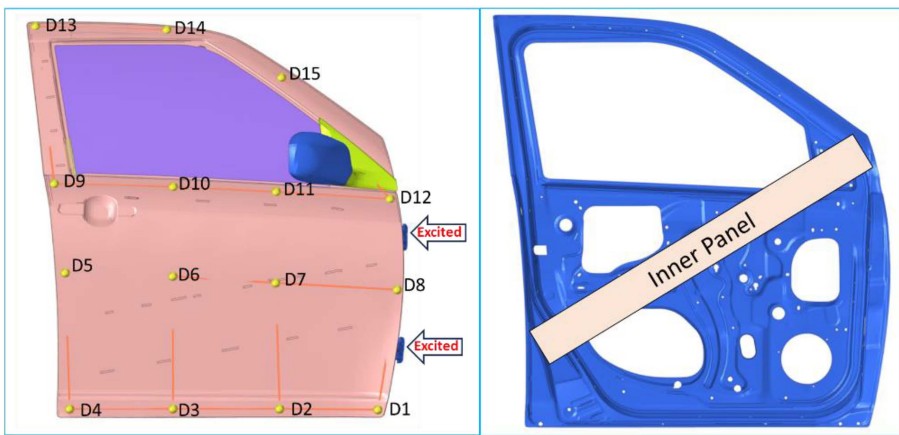

**Fig 4.  Door Assembly Critical Location and Inner Panel.**

displacement responses were taken in all three XYZ directions and converted into stiffness. Since the stiffness in the Y direction is more noticeable than in the other two, only the Y direction's stiffness was used for the final evaluation, which is perpendicular to door surfaces. For the frequency range of 20 Hz to 100 Hz, the minimum stiffness of each crucial region has been calculated and compared to the designated thickness optimization target. According to the stipulated aim, the local stiffness perpendicular to the outer surface of the door must be larger than 8.5 N/m at all critical locations/ key points, which, in turn, indicates throughout the door's outer surface.

### 3.3. Acceleration results evaluation method

Like the dynamic stiffness analysis, to assess the acceleration, the specified 15 critical locations have been taken. All points were excited at a time by applying excitations at hinge locations. The acceleration results for the excitation of both hinge locations are similar; therefore, the findings from the bottom hinge location have been selected for further analysis and presentation as a representative example. All eight models' results from 0 Hz to 100 Hz have been calculated using the Altair OptiStruct solver at all 15 critical locations. The acceleration outcomes in the Y direction, which are perpendicular to the outer panel, are significant and distinctly observable. The maximum acceleration at each critical location has been recorded for the frequency range of 20 Hz to 100 Hz, and these values have been compared against one another, as well as the designated targets for thickness optimization. The aim is to ensure that the acceleration perpendicular to the outer panel of the door remains below 3.0 m/sec$^2$ at all critical points, which effectively translates to maintaining this limit across the entire surface of the door's outer panel.

## 4. Finite element model of door assembly

The complete door assembly's finite element (FE) model was created based on the 3D CAD model, which was constructed using scanned data from the individual door components. The baseline thickness for the door FE mode, set at T14 = 1.367 mm, was optimized through a sensitivity analysis of various parameters conducted in prior research. The baseline finite element (FE) model, designated as T14, served as a reference point for the development of seven additional models, each with varying thicknesses: 1.0 mm (T10), 1.2 mm (T12), 1.6 mm (T16), 1.8 mm (T18), 2.0 mm (T20), 2.2 mm (T22), and 2.4 mm (T24).

All components were discretized using 2D shell elements and 3D hexahedral elements. The panels were meshed with 2D shell elements measuring 10 mm at their midpoint, incorporating the appropriate thicknesses [31], while 3D hexahedral elements were employed for the hinge brackets. All minor components, such as the handle, speaker, latch, and switch box, were represented using RBE3-Com2, incorporating their measured weights (Fig 5). Weather seals are represented using RBE3-CBush-RBE3 configurations, characterized by bush properties of K1: 15, K2: 78 N/mm, and K3: 15 N/mm.

The connections for both hinges were established using RBE2, allowing the hinge pin rotation to remain unrestricted. RBE3-Hex-RBE3 elements are utilized to model the adhesive and spot connections, incorporating the corresponding material properties of the 3D hexagonal element. The seam weld connection was represented using 2D shell elements between two steel panels, with these elements reflecting the minimum thickness of the connection panels. Structural parts such as the inner panel, outer panel, stiffeners, glass rails and regulator mechanism, and steel material were used. The regulator motor model material was aluminum, the glass had glass material, and the trim parts were plastic. Details about the material used are given in Table 2 below.

### 4.1. Door assembly finite element analysis

The modal and frequency response analysis (FRA) of all eight-door assembly models is examined within the relevant frequency range of 0 Hz to 100 Hz in relation to the vibrations. In the Opti-Struct solver, the modal analysis is carried out under free-free conditions between 0 Hz to 100 Hz, and the frequencies and mode shapes are computed. Listed are pure global bending modes according to the optimization needs. The 25 Hz to 60 Hz frequency range contains all pure bending

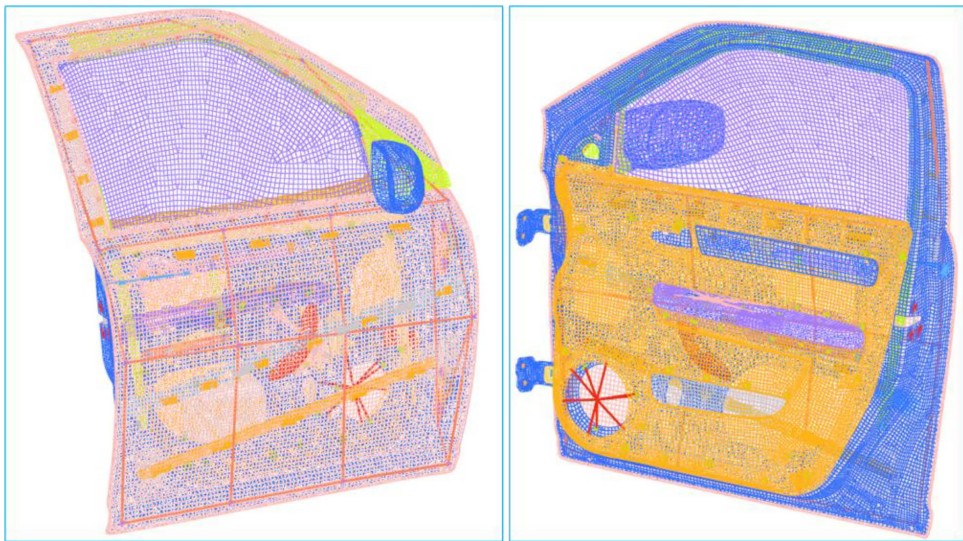

**Fig 5. FE Model of the Door Assembly.**

**Table 2. List of Material Properties Used for FE Model.**

| Material | Modulus of Elasticity/ Speed of Sound | Unit | Poisons ratio | Density (Tones/mm³) |
|---|---|---|---|---|
| Steel | $2.1 \times 10^5$ | N/mm² | 0.29 | $7.85 \times 10^{-9}$ |
| Plastic | $1.1 \times 10^3$ | N/mm² | 0.3 | $1.0 \times 10^{-9}$ |
| Glass | $6.5 \times 10^4$ | N/mm² | 0.24 | $2.4 \times 10^{-9}$ |
| Adhesive | 1.9 | N/mm² | 0.36 | $1.2 \times 10^{-9}$ |
| Isolator/ Dampener | 4.51 | N/mm² | 0.48 | $1.24 \times 10^{-9}$ |
| Air – Cavity | $3.4 \times 10^5$ | mm/sec | – | $1.20 \times 10^{-12}$ |

modes. The bending behavior of modes above 60 Hz is complicated and a little challenging to identify. Which will make vibration and vibrational resonance more likely. All of the modes in the frequency range of 0 Hz to 100 Hz are reviewed, and the pure directional bending modes are listed and compared with the frequency data from the modal test. Table 3 lists the frequency values of modal tests and the associated bending behavior.

Consider the frequency range of 0 Hz to 100 Hz, which is based on the natural frequencies of the human body and optimization guidelines. From 0 Hz to 100 Hz, modal and frequency response analyses are conducted. The dynamic

**Table 3. Experimental Test and FE Analysis Frequency (Hz) Comparison. (Door Assembly (26.7 kg): Inner Panel Thickness Effect).**

| Modal Test – Mode Shape Description | Modal Test | T10 | T12 | T14 | T16 | T18 | T20 | T22 | T24 | Optimized T15 = 1.52 |
|---|---|---|---|---|---|---|---|---|---|---|
| **Mass (kg)** | **26.6** | **24.12** | **25.53** | **26.70** | **28.34** | **29.74** | **31.15** | **32.56** | **33.96** | **27.78** |
| 1st Lateral Bending | 26.2 | 25.9 | 26.0 | 26.1 | 26.2 | 26.3 | 26.4 | 26.4 | 26.5 | 26.2 |
| 1st Vertical Bending | 32.4 | 35.3 | 35.7 | 36.0 | 36.2 | 36.4 | 36.5 | 36.6 | 36.7 | 36.1 |
| 2nd Vertical Bending | 38.6 | 39.0 | 40.1 | 40.8 | 41.4 | 41.8 | 42.1 | 42.3 | 42.4 | 41.3 |
| 2nd Lateral Bending | 43.5 | 39.8 | 41.6 | 42.5 | 43.5 | 44.3 | 44.9 | 45.5 | 46.1 | 43.2 |
| 1st Torsional Bending | 49.4 | 49.7 | 50.5 | 51.0 | 51.6 | 52.0 | 52.6 | 53.4 | 54.4 | 51.4 |
| 2nd Torsional Bending | 55.4 | 52.6 | 54.5 | 56.0 | 58.4 | 58.5 | 60.4 | 61.3 | 61.5 | 57.1 |

stiffness and vibration frequency response functions (FRF) are computed for each of the 15 critical locations (Fig 4). In the FRF analysis case of a critical location, both hinge locations are excited in all XYZ directions. On the door's outer panel, the dynamic stiffness and vibration results are examined. The amplitude of the dynamic stiffness and vibration response in the Y direction, or perpendicular to the door outer panel, is noticeably high and easy to understand. At both hinge locations, the vibrations for the Y direction excitation are similar; however, the vibrations at the bottom hinge are higher than those at the top. All 15 critical locations are further evaluated using the vibration results of bottom excitation.

### 4.2. Door Assembly Modal Testing

A Maruti Suzuki Swift Dzire car's baseline door assembly (T14 = 1.367 mm) was tested in the ARAI NVH Lab using the LMS-Siemens (Scadas & Test Lab) system (Figs 4 and 6b). The door assembly was suspended on bungee cords as a standard procedure. The outer surface of the door assembly was spatially divided into 15 areas (D1, to D15) that were chosen as critical response locations. To determine the acceleration at these fifteen points, excitations were applied to both hinges, top (D16) and bottom (D17). For the acceleration FRF calculation, a door was excited by an impact hammer of 500 lbf (the PCB made) in all three directions at both hinge locations. To determine the displacement, the same hammer was used to independently excite each of the fifteen locations. Surface vibrations and displacements between the frequency range of 0 Hz to 500 Hz were measured using 15 triaxial accelerometers (PCB -356A01 & Dytran made ~4.2 gm).

## 5. Post processing results and discussion

### 5.1. Modal analysis results

To determine the modal frequencies and associated mode shape, a free-free modal analysis of the baseline (T14 = 1.367 mm) door assembly (26.7 kg) is conducted. Based on the frequency range of typical, healthy human body parts, the analysis frequency range is chosen. The frequency range for human body parts is 0 Hz to 100 Hz.

In Table 3 below and the graph (Fig 7), the global bending modes of the door assemblies for each of the eight models (T10, T12, T14, T16, T18, T20, T22, and T24) are listed and compared with experimental modal frequencies. Here, in the final column, are the modal frequencies of the optimized model (T15 = 1.52 mm). The minimum dynamic stiffness and maximum vibration of each of the 15 critical locations have been used to determine an optimal thickness model.

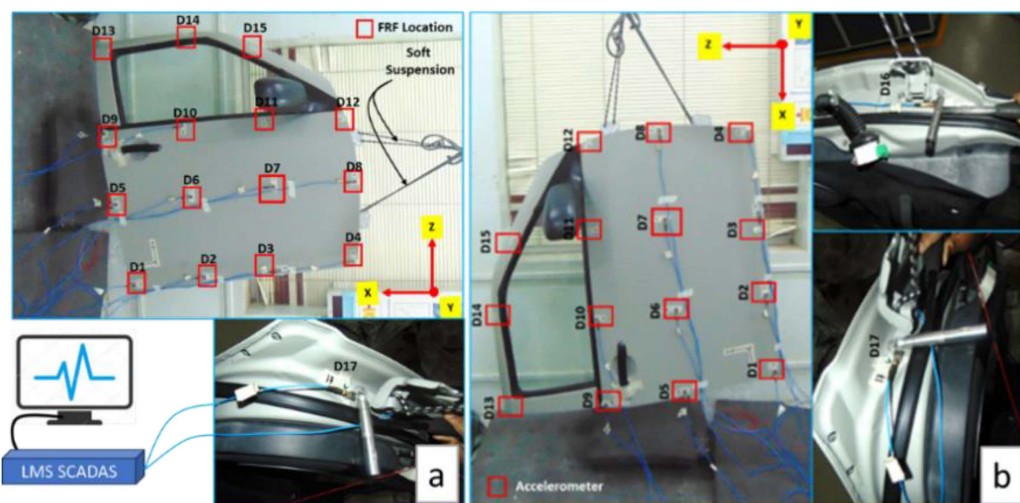

**Fig 6. Door Test Set-up (a) Schematic and (b) Experimental Test Set-up.**

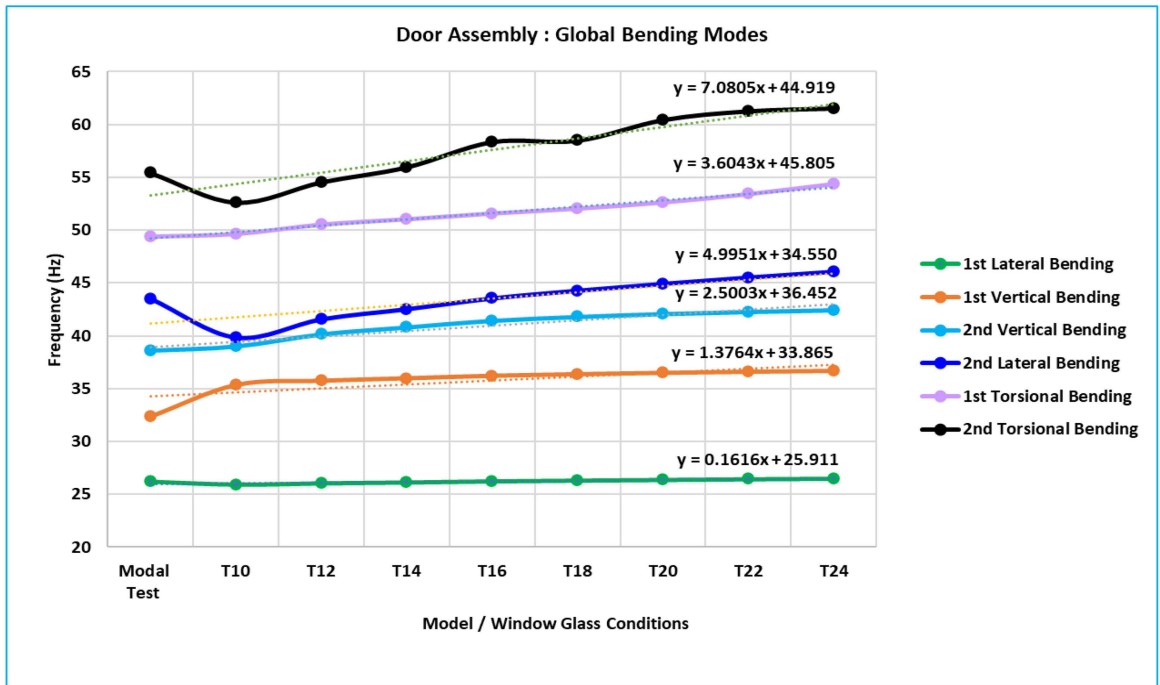

**Fig 7. Door Assembly Global Bending Mode Frequency with Trend Line Equation.**

Below is a summary of the variations in inner panel thickness: Table 3 and Fig 7 above show the frequencies of global bending modes of all eight models with a thickness-optimized model.

- The frequencies of experimental tests are comparable to all global bending modes.

- The global bending mode frequencies also change (increase or decrease) in proportion to the inner panel's thickness changes.

- The thickness of the inner panel has an approximately linear relationship with each of the six global bending modes.

- This has led to the establishment of a linear trend line equation for every bending mode [9], which will be utilized to determine the unknown parameter, such as the inner panel's thickness or the frequency of any bending mode.

### 5.2. Dynamic stiffness results of all critical locations

As was indicated in the previous section, the critical or concerned frequency range is between 0 Hz to 100 Hz; however, in all eight models, the door assembly's natural modes begin at about 24 Hz. Thus, the vibration's frequency response function (FRF) in relation to dynamic stiffness is plotted between 20 and 100 Hz. The dynamic stiffnesses, the FRF curves of all eight models (T10 to T24) of the corresponding locations overlap with the FRFs of all 15 critical locations (D1 to D15). The door assembly's natural/global modes are indicated by these dynamic stiffness FRFs. When the eight curves in a graph are overlapped, a distinct pattern of all curves with variations in inner panel thickness is visible (Figs 8 and 11).

D1, D2, D3, and D4 are the door's bottommost critical locations (Fig 8). Stiffness increases with increase of thickness across the whole 0 Hz to 100 Hz frequency range. The natural frequencies of the door assembly clearly show an increase in stiffness amplitude.

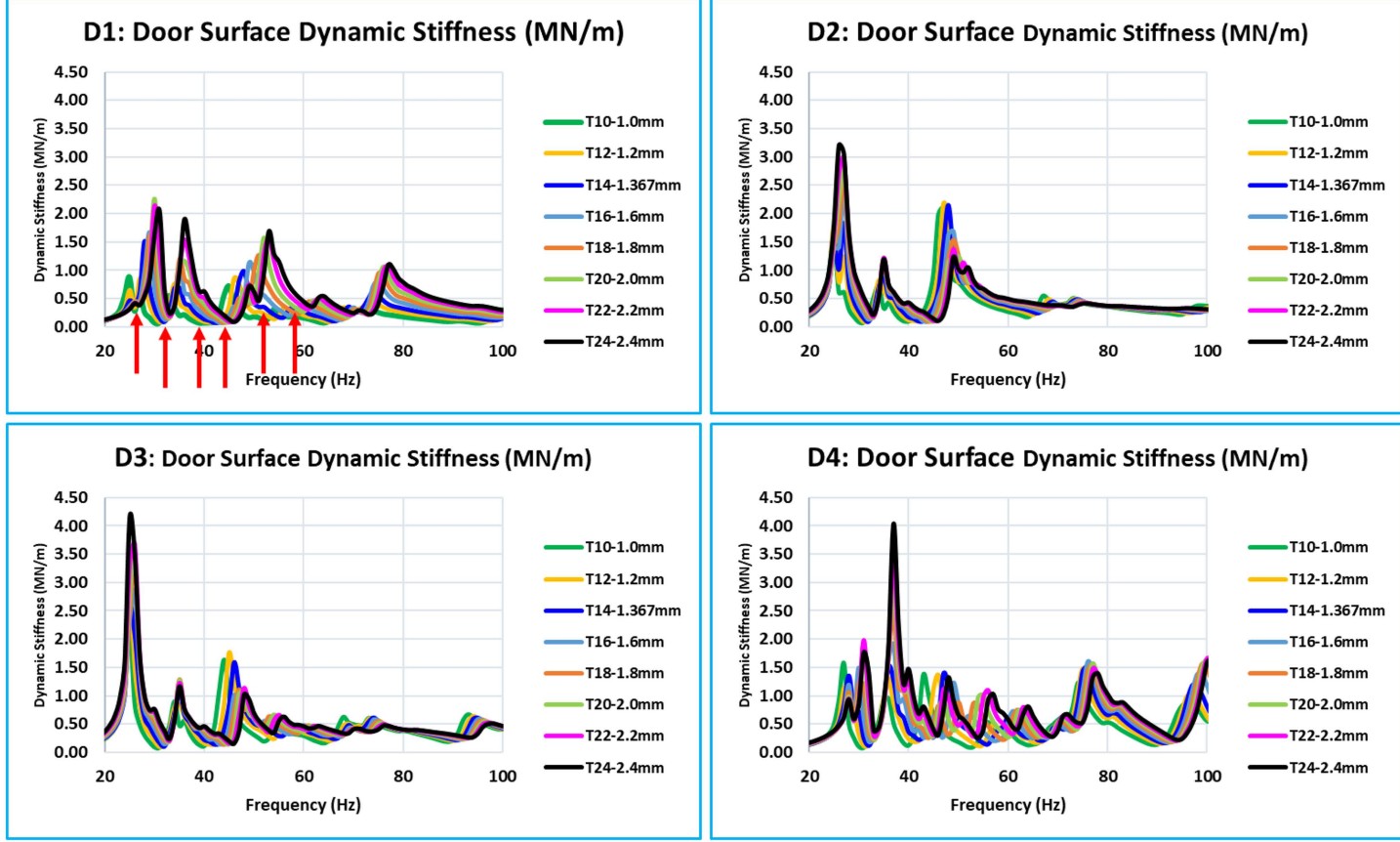

**Fig 8. D1, D2, D3, and D4 - Dynamic Stiffness Comparison.**

D5, D6, D7, and D8 are critical locations in the center of the door plane area (Fig 9). Every location of this level has a unique stiffness curve pattern between 0 Hz to 100 Hz. The trend of increasing stiffness is directly proportional to the thickness of the inner panel. Since D6 and D7 are critical locations in the middle of the door and represent the absolute stiffness of the outer panel, their levels of stiffness are much lower than those of D5 and D8.

The bottom of the glass window is where the critical locations (Fig 10) are located: D9, D10, D11, and D12. In this case, the relationship between increasing thickness and increasing stiffness is the same across the whole frequency range of 0 Hz to 100 Hz. Compared to D10 and D11, the stiffness level of D9 and D12 is lower. These critical locations are in the center of the door window, and because of the window cutout, they have less support.

The glass window's uppermost D13, D14, and D15 are the critical locations (Fig 11). For these locations, for the frequency range of 0 Hz to 100 Hz, the relationship between thickness and stiffness is the same. These three places have a low amplitude of stiffness in comparison to other critical locations. These places are on the uppermost portion of the window frame structure, which is a thin structural part.

### 5.3. Overall summary of stiffness results of 15 critical locations -

Plots of the dynamic stiffness FRFs for each of the 15 locations (D1 to D15) are shown in Figs 8 to 11. They have clear indications about the global pure bending and complex bending modes of the door. The following are the interpretations

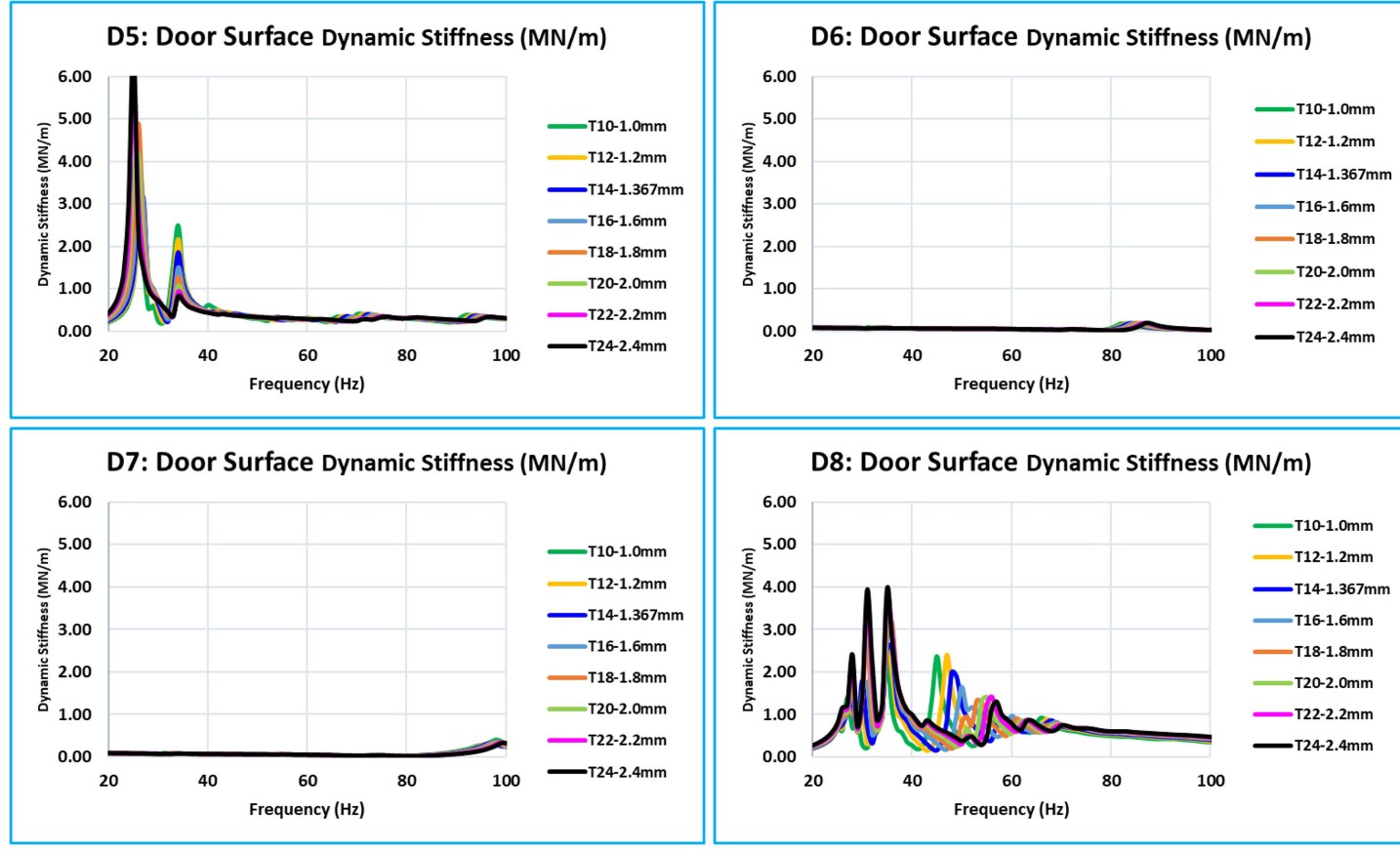

**Fig 9. D5, D6, D7, and D8 - Dynamic Stiffness Comparison.**

based on an examination and comparison of all 15 locations where the curves of all eight models of different thicknesses overlap.

- The dynamic stiffness at all critical locations follows a similar pattern, with thickness variation ranging from 1.0 mm to 2.4 mm (T10 to T24 models). The fluctuation in dynamic stiffness deep or weakness is roughly linear for the entire frequency range of concern with thickness variation.

- Fig 8 of the D1 location curve shows the red arrow, which indicates that the majority of the critical locations have dynamic stiffness curves with deep or relatively low dynamic stiffness values at global modes.

- Table 4 lists the minimum dynamic stiffness of all 15 critical locations for all eight models (T10 to T24) between 20 Hz and 60 Hz. The desired value of 8.5 N/m and these minimal dynamic stiffness values are compared across all 15 important places.

- For each of the 15 critical locations of thickness (T10 to T24), the dynamic stiffness value filled in with yellow is the lowest value.

- The values of models T14 = 7.533 N/m and T16 = 11.188 N/m are the closest to the desired value of 8.5 N/m among all of these minimum dynamic stiffness values. The findings of this linear analysis exhibit a linear relationship because of the

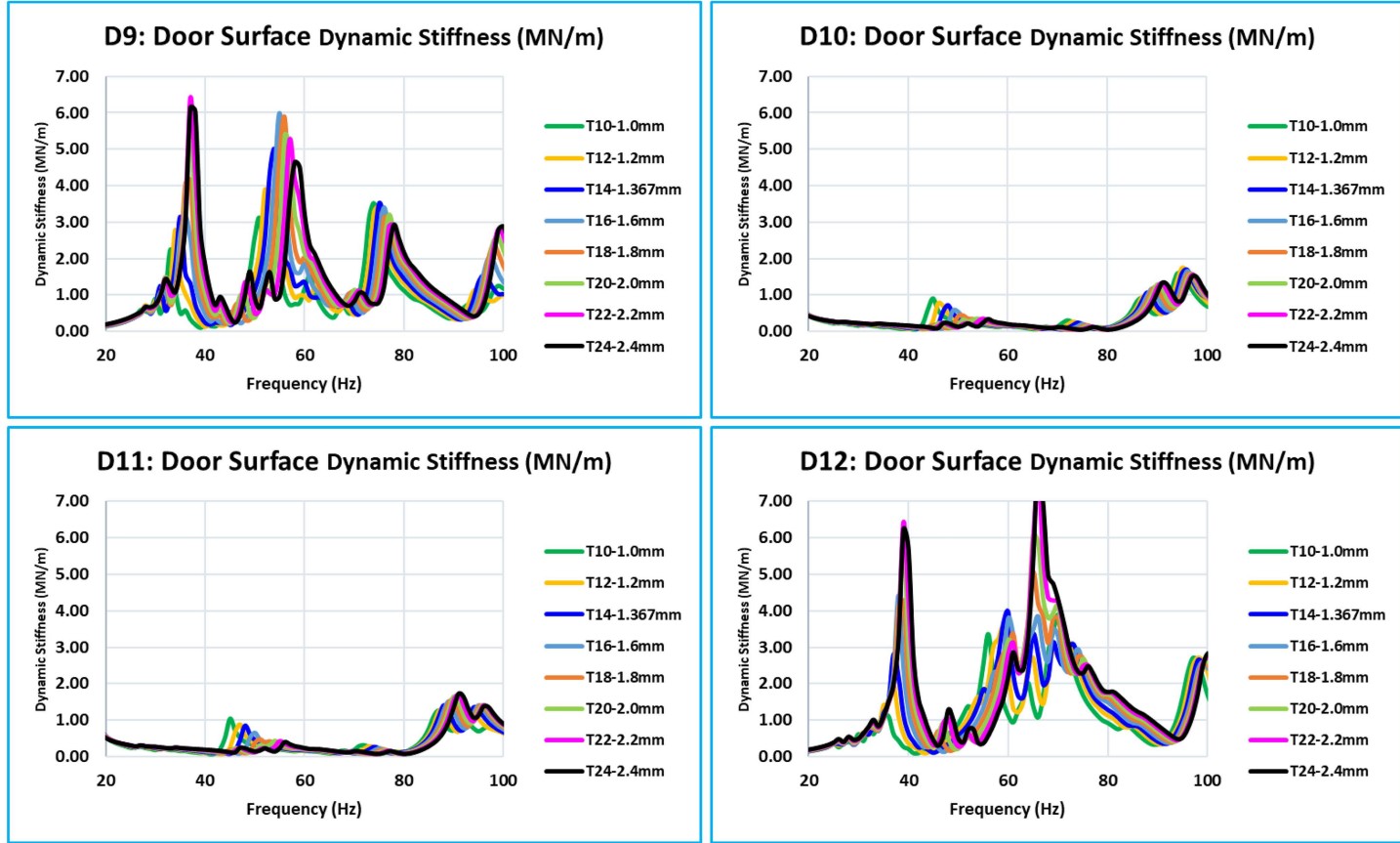

**Fig 10. D9, D10, D11, and D12 - Dynamic Stiffness Comparison.**

thickness variations. The optimum thickness is determined to satisfy the dynamic stiffness using equation (6) and the linear interpolation approach [10]. The optimum calculated value is 1.429 mm (equation 9)

Where, $x_0$ = 7.533 N/m, $x_1$ = 11.188 N/m, x = 8.5 N/m

$$y_0 = 1.367 \text{ mm}, \quad y_1 = 1.600 \text{ mm}, \quad y = ? \text{ mm}$$

$$y = 1.367 + \frac{(8.5 - 7.533)(1.6 - 1.367)}{(11.188 - 7.533)} \tag{7}$$

$$y = 1.367 + 0.061645 \tag{8}$$

$$y = 1.429 \text{ mm} \tag{9}$$

- Based on the dynamic stiffness, a calculated thickness value is 1.429 mm. Ultimately, the optimum thickness value will be chosen by taking into account the calculated values for dynamic stiffness and vibration, as well as the standard sheet

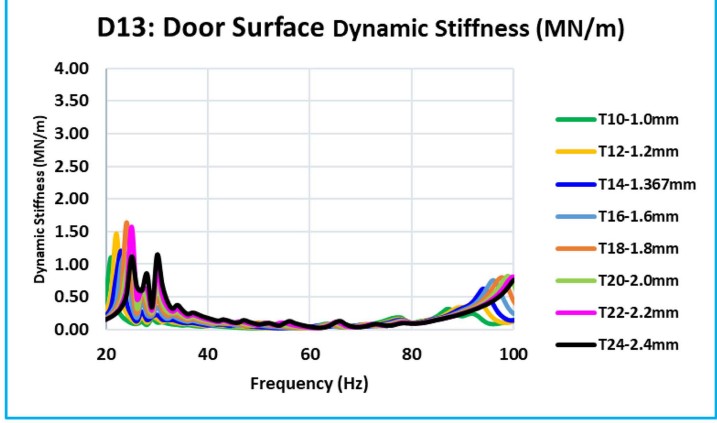

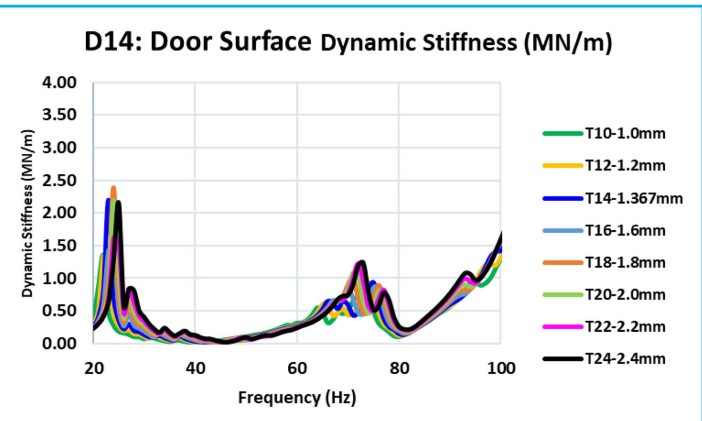

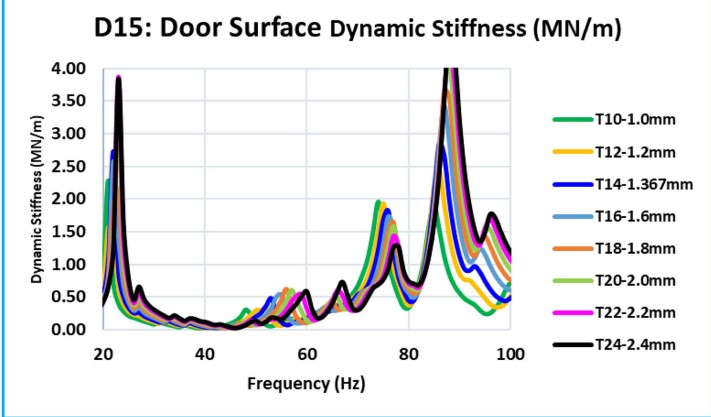

**Fig 11. D13, D14, and D15 - Dynamic Stiffness Comparison.**

**Table 4. Minimum Dynamic Stiffness at Critical Locations Between 20 Hz – 60 Hz.**

| Minimum Dynamic Stiffness (N/m) at Critical Locations between 20 Hz – 60 Hz | | | | | | | | | | | | | | |
|---|---|---|---|---|---|---|---|---|---|---|---|---|---|---|
| Model | D1 | D2 | D3 | D4 | D5 | D6 | D7 | D8 | D9 | D10 | D11 | D12 | D13 | D14 | D15 |
| T10-1.0 mm | 70.70 | 75.32 | 80.27 | 77.60 | 200.81 | 42.87 | 35.29 | 176.22 | 108.73 | 63.72 | 54.69 | 84.84 | 5.65 | 15.39 | 16.92 |
| T12-1.2 mm | 73.53 | 85.34 | 91.59 | 108.81 | 219.25 | 44.34 | 36.38 | 137.72 | 137.41 | 66.04 | 56.20 | 86.62 | 6.91 | 17.81 | 19.26 |
| T14-1.367 mm | 69.83 | 91.22 | 101.44 | 110.88 | 233.77 | 45.43 | 37.49 | 148.40 | 161.38 | 78.90 | 67.31 | 95.72 | 7.53 | 18.31 | 24.63 |
| T16-1.6 mm | 72.58 | 95.57 | 138.26 | 139.09 | 263.69 | 46.71 | 38.87 | 167.52 | 169.91 | 85.76 | 76.56 | 119.21 | 11.19 | 16.71 | 22.60 |
| T18-1.8 mm | 68.29 | 87.14 | 131.35 | 144.91 | 267.09 | 47.59 | 39.91 | 189.06 | 177.75 | 83.54 | 81.65 | 161.05 | 12.76 | 15.08 | 19.78 |
| T20-2.0 mm | 73.02 | 92.39 | 143.90 | 151.27 | 269.66 | 48.23 | 40.89 | 233.90 | 185.93 | 95.58 | 94.22 | 168.20 | 14.78 | 14.51 | 19.02 |
| T22-2.2 mm | 87.42 | 104.70 | 150.75 | 158.02 | 275.67 | 48.90 | 41.88 | 248.12 | 194.40 | 88.87 | 95.18 | 175.55 | 24.97 | 17.85 | 22.82 |
| T24-2.4 mm | 83.65 | 101.70 | 151.82 | 165.08 | 283.44 | 49.55 | 42.87 | 261.97 | 203.10 | 96.08 | 104.53 | 183.08 | 39.92 | 15.57 | 19.95 |
| T15-1.52 mm | 62.09 | 81.91 | 117.75 | 136.95 | 246.82 | 46.30 | 38.42 | 153.75 | 166.90 | 81.71 | 70.55 | 121.30 | 9.26 | 15.04 | 20.08 |

thickness (gauge) that is readily available in the market, which is close to the calculated value. This computed thickness value is close to 1.52 mm thick (gauge 16). The dynamic stiffness results for an inner panel that is 1.52 mm thick are displayed in Fig 12 below (in cyan color).

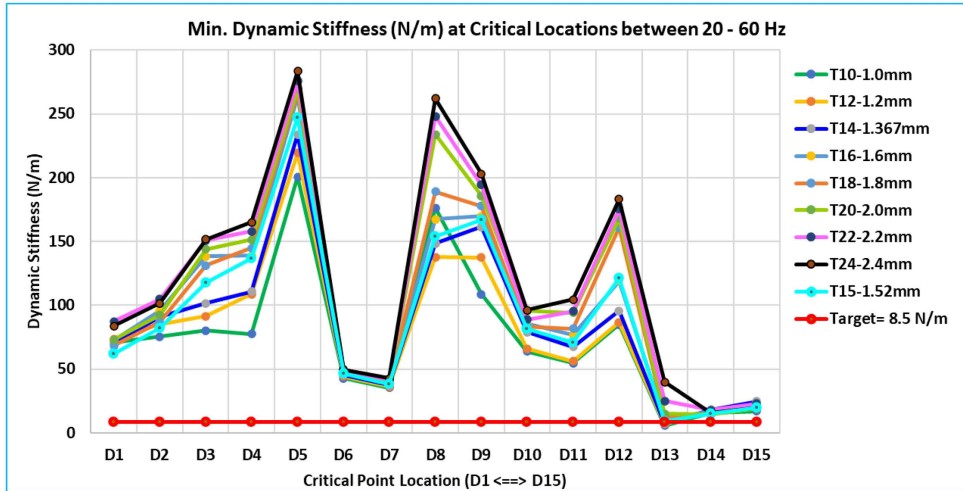

**Fig 12. Minimum Dynamic Stiffness – Critical Location between 20–60 Hz.**

## 5.4. Vibration results of all critical locations

The bottom of the door has critical locations D1, D2, D3, and D4 (Fig 13), all of which exhibit vibration peaks at the door assembly's natural frequencies. The vibrations decrease over the whole frequency range of 0 Hz to 100 Hz as the inner panel's thickness increases. As the thickness of the inner panel increases, there is a noticeable decrease in the vibration amplitude at the door assembly's natural frequencies.

These critical locations, D5, D6, D7, and D8, are situated in the center of the door plane area (Fig 14). At this level, every location has a unique vibration curve pattern between 0 Hz to 100 Hz. The trend toward less vibration is closely correlated with the thickness of the inner panel.

Critical locations D9, D10, D11, and D12 are located at the bottom of the glass pane (Fig 15). In this case, the relationship between thickness increases and vibration reduction is the same across the whole 0 Hz to 100 Hz frequency range. With an increase in thickness, there is definite evidence of a greater reduction in vibration amplitude at the door assembly's natural frequencies in comparison to other frequencies.

The top of the glass window contains the critical locations (D13, D14, and D15) (Fig 16). Additionally, the relationship between the increase in thickness and decrease in vibration is the same at frequencies ranging from 0 to 100 Hz. Compared to every other critical location on the door, D13 has a high vibration intensity. These locations are on top of the window frame structure, where a narrow piece of the structure is present.

## 5.5. Overall summary of acceleration results of 15 critical locations -

Figs 13 to 16 plot and display the vibration FRFs for each of the 15 locations (D1 to D15). The global bending modes and complex bending of the door are clearly indicated by them. The door's global mode is indicated by the noticeably high vibrations; maximal vibrations occur at natural frequencies. The interpretations are presented below following the examination and interpretation of all 15 places and overlapping curves of all eight models of different thicknesses.

- With thickness variations ranging from 1.0 mm to 2.4 mm (T10 to T24 models), vibration FRFs for all critical locations exhibit comparable patterns, and vibration peaks or amplitudes vary roughly linearly. Vibration reduces over the whole frequency range in question (0 Hz to 100 Hz) as thickness increases.

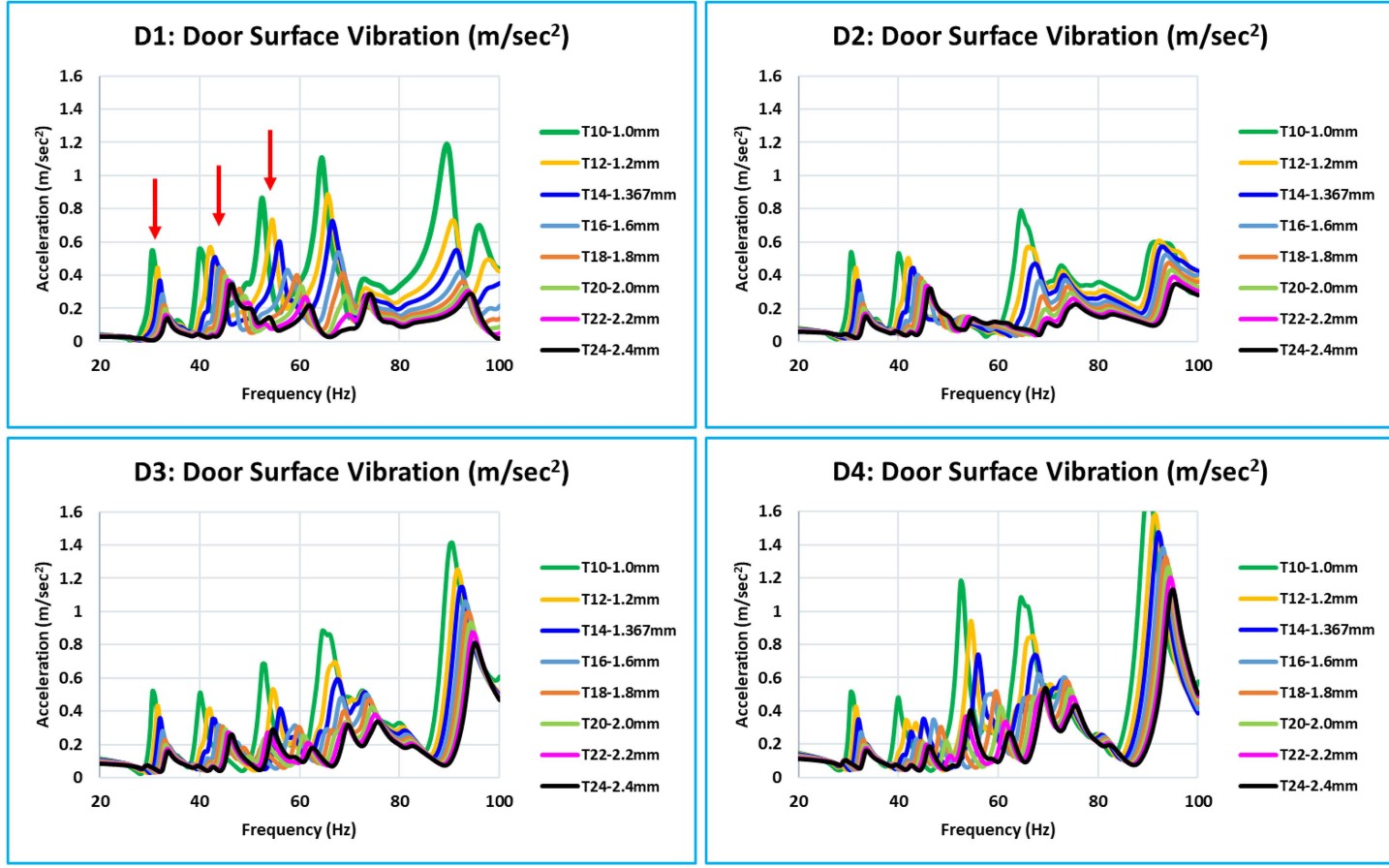

**Fig 13. D1, D2, D3, and D4 - Vibration Comparison.**

- In Fig 13 of the D1 location curve, the red arrow indicates that the majority of the vibration FRF curve at the critical location has a peak or relatively high amplitude at global modes.

- Table 5 lists the maximum vibration of all 15 critical locations for all eight models (T10 to T24) between 0 Hz to 100 Hz. These maximum vibration values are compared to a target value of 3.0 m/sec² at each of the 15 critical locations.

- The vibration value that is indicated or filled in with yellow is the highest value among the 15 critical locations of corresponding thickness for each model (T10 to T24).

- Values of models T14 = 3.219 m/sec² and T16 = 2.312 m/sec² are the closest to the desired value of 3.0 m/sec² among all of these maximum vibration values. Vibration results show a linear relationship with thickness variations because this is a linear study. The optimum thickness to achieve the vibration target value of 3.0 m/sec² is determined using equation (6) and the linear interpolation approach [10]. The optimum calculated value is 1.423 mm (equation 12).

where, $x_0$ = 3.219 m/sec², $x_1$ = 2.312 m/sec², $x$ = 3.0 m/sec²

$$y_0 = 1.367 \text{ mm}, \quad y_1 = 1.600 \text{ mm}, \quad y = ? \text{ mm}$$

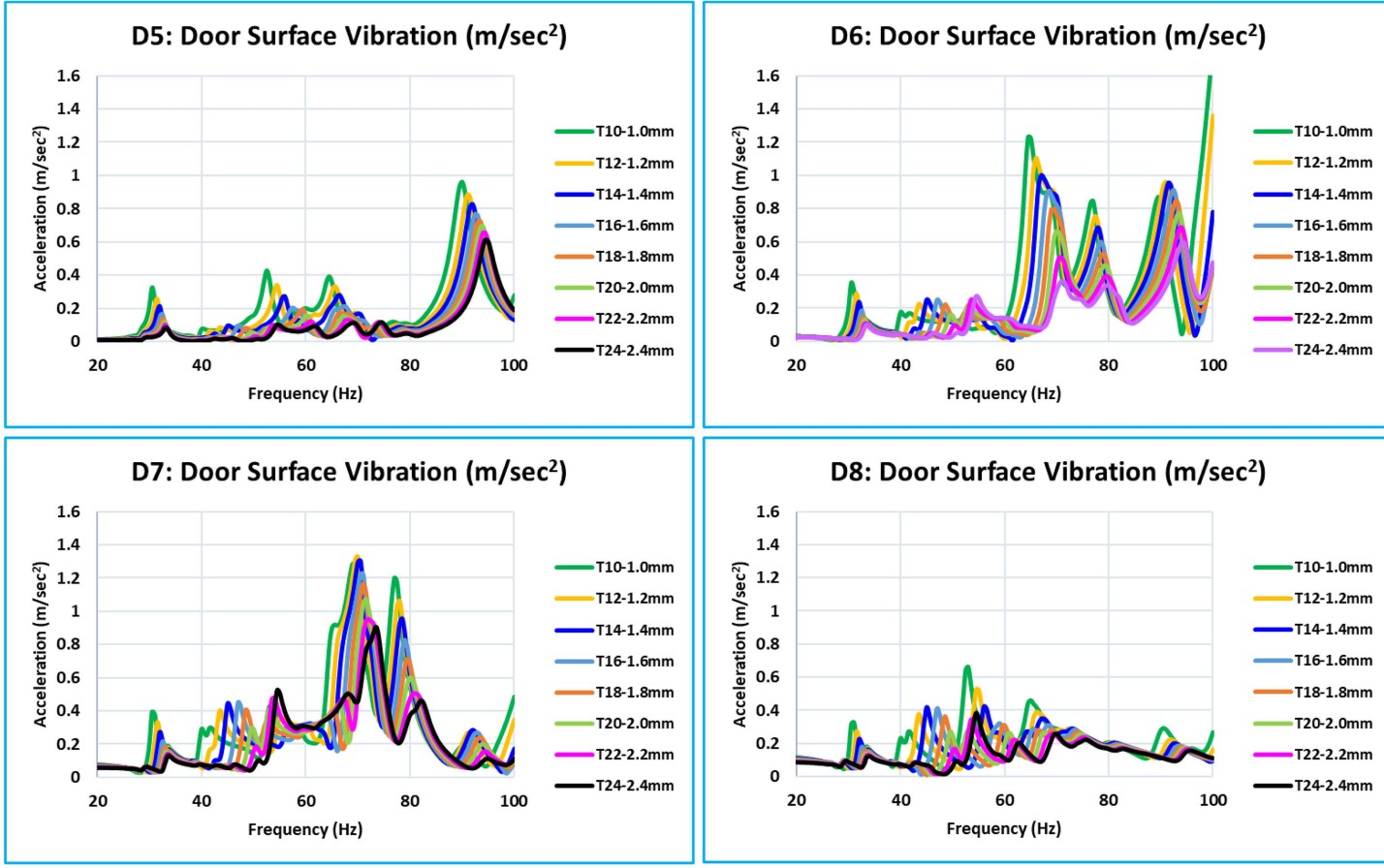

**Fig 14. D5, D6, D7, and D8 - Vibration Comparison.**

$$y = 1.367 + \frac{(3.0 - 3.219)(1.6 - 1.367)}{(2.312 - 3.219)} \tag{10}$$

$$y = 1.367 + 0.056259 \tag{11}$$

$$y = 1.423 \tag{12}$$

- Based on the vibration, a calculated thickness value is 1.423 mm. Ultimately, the optimum thickness value will be chosen by taking into account the calculated values for dynamic stiffness and vibration, as well as the standard sheet thickness (gauge) that is readily available in the market, which is close to the calculated value. This computed thickness value is close to 1.52 mm thick (gauge 16).

- In the current baseline model (T14), the inner panel should have a thickness of 1.52 mm rather than 1.367 mm, to satisfy the specified target values. The vibration results for an inner panel that is 1.52 mm thick are displayed in Fig 17 below (in cyan color).

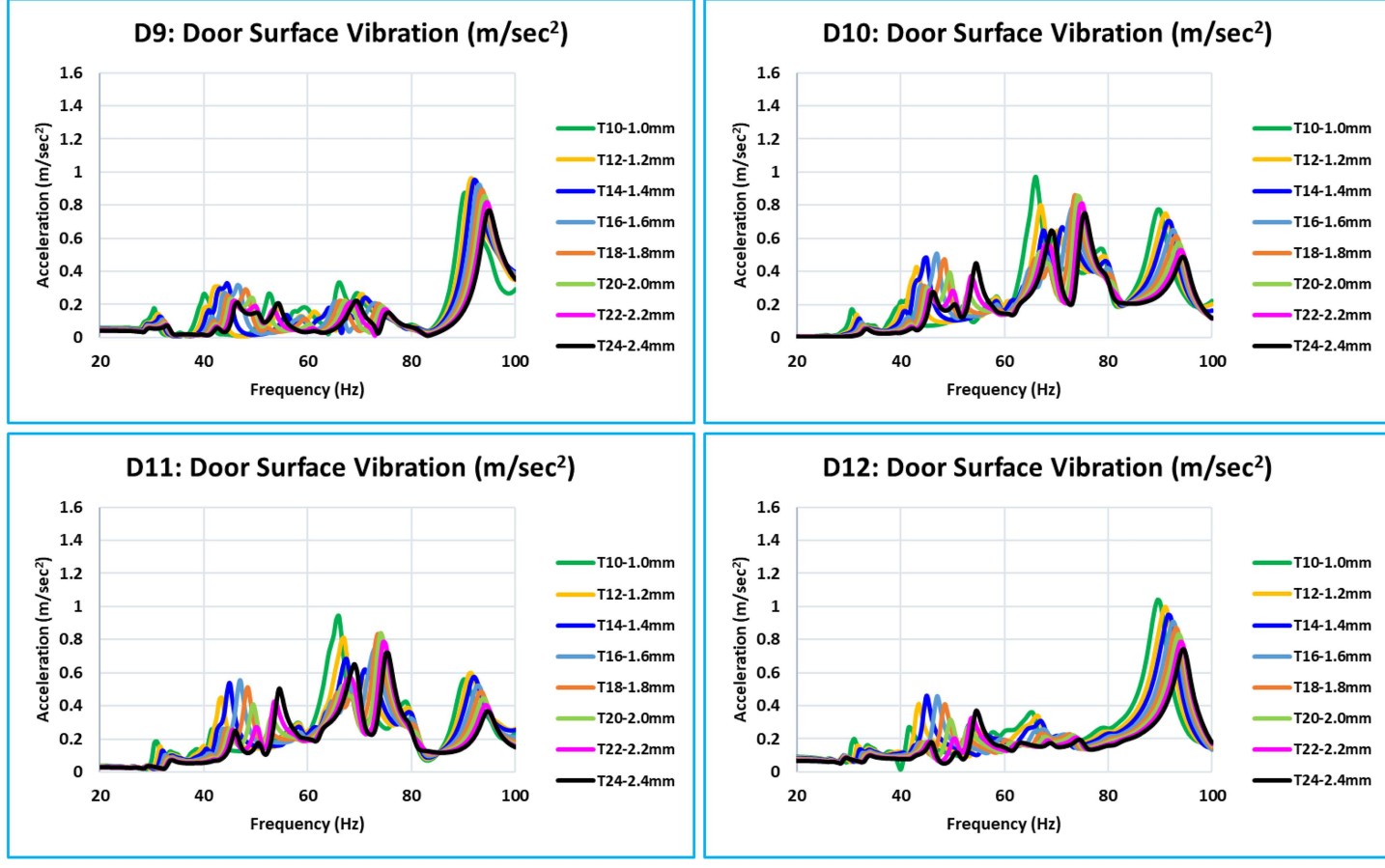

**Fig 15. D9, D10, D11, and D12 - Vibration Comparison.**

## 6. Conclusions

All the results of the door assembly analysis with thickness variations are tabulated in Tables 3–5. After reviewing and interpreting these results, the optimized model was derived as T15 = 1.52 mm. The final results of the optimized and baseline models are compared in Table 6. Based on all these results of modal analysis, stiffness analysis, and vibration (acceleration) analysis, the following top-level conclusions cum relations have been established.

- The thickness variation of the inner panel has a direct correlation with the frequency of the door assembly's global bending modes.

- The thickness variation of the inner panel has an inverse relationship with the dynamic stiffness of the door assembly.

- The thickness variation of the inner panel directly correlates with the vibrations of the door assembly's critical locations.

- The door assembly's top left corner, critical location D13, has the lowest dynamic stiffness and the highest vibrations; this indicates that dynamic stiffness and vibration are opposite to each other.

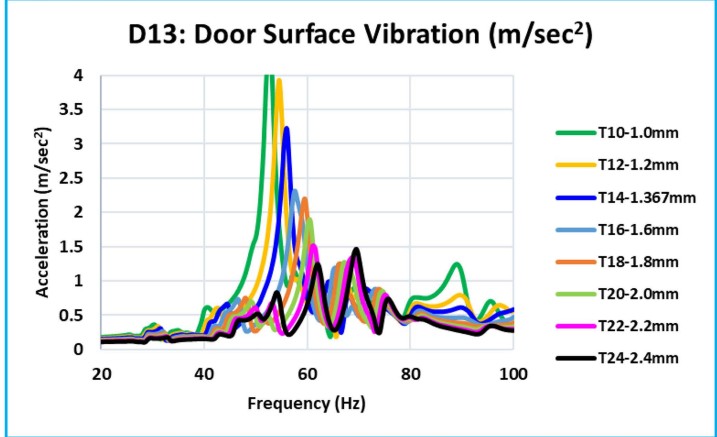

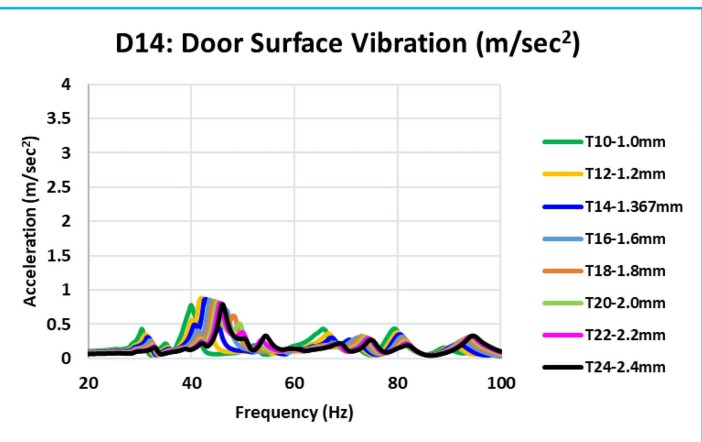

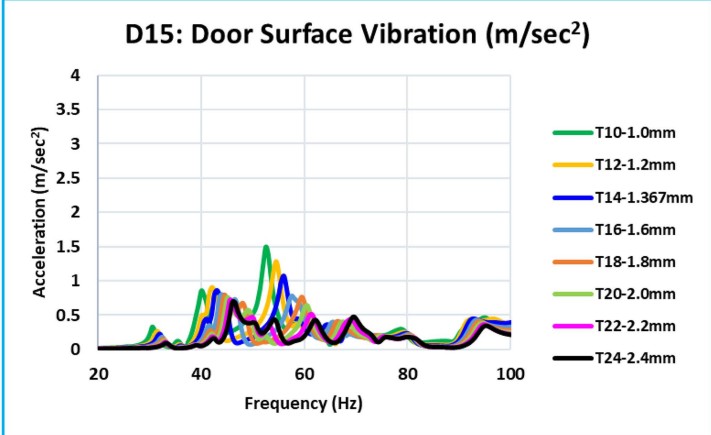

**Fig 16. D13, D14, and D15 - Vibration Comparison.**

**Table 5. Maximum Vibration at Critical Locations Between 0 Hz – 100 Hz.**

Maximum Vibrations (m/sec²) at Critical Locations between 0 Hz – 100 Hz

| Model | D1 | D2 | D3 | D4 | D5 | D6 | D7 | D8 | D9 | D10 | D11 | D12 | D13 | D14 | D15 |
|---|---|---|---|---|---|---|---|---|---|---|---|---|---|---|---|
| T10-1.0 mm | 1.192 | 0.787 | 1.412 | 1.787 | 0.962 | 1.809 | 1.287 | 0.664 | 0.876 | 0.972 | 0.945 | 1.043 | 4.660 | 0.775 | 1.499 |
| T12-1.2 mm | 0.884 | 0.606 | 1.253 | 1.580 | 0.885 | 1.361 | 1.326 | 0.526 | 0.966 | 0.800 | 0.808 | 0.997 | 3.925 | 0.887 | 1.284 |
| T14-1.367 mm | 0.725 | 0.572 | 1.145 | 1.476 | 0.828 | 0.999 | 1.306 | 0.421 | 0.951 | 0.702 | 0.681 | 0.949 | 3.219 | 0.859 | 1.067 |
| T16-1.6 mm | 0.536 | 0.522 | 1.058 | 1.383 | 0.766 | 0.911 | 1.224 | 0.412 | 0.923 | 0.788 | 0.750 | 0.909 | 2.312 | 0.843 | 0.800 |
| T18-1.8 mm | 0.426 | 0.477 | 0.992 | 1.326 | 0.725 | 0.850 | 1.162 | 0.365 | 0.894 | 0.858 | 0.834 | 0.871 | 2.196 | 0.840 | 0.789 |
| T20-2.0 mm | 0.394 | 0.432 | 0.932 | 1.267 | 0.690 | 0.773 | 1.075 | 0.281 | 0.859 | 0.852 | 0.837 | 0.831 | 1.887 | 0.834 | 0.758 |
| T22-2.2 mm | 0.363 | 0.389 | 0.873 | 1.202 | 0.651 | 0.688 | 0.953 | 0.344 | 0.818 | 0.812 | 0.786 | 0.788 | 1.518 | 0.811 | 0.722 |
| T24-2.4 mm | 0.346 | 0.348 | 0.813 | 1.130 | 0.616 | 0.597 | 0.905 | 0.386 | 0.770 | 0.754 | 0.719 | 0.743 | 1.463 | 0.785 | 0.691 |
| T15-1.52 mm | 0.592 | 0.538 | 1.084 | 1.408 | 0.788 | 0.938 | 1.268 | 0.419 | 0.929 | 0.739 | 0.690 | 0.915 | 2.622 | 0.864 | 0.879 |

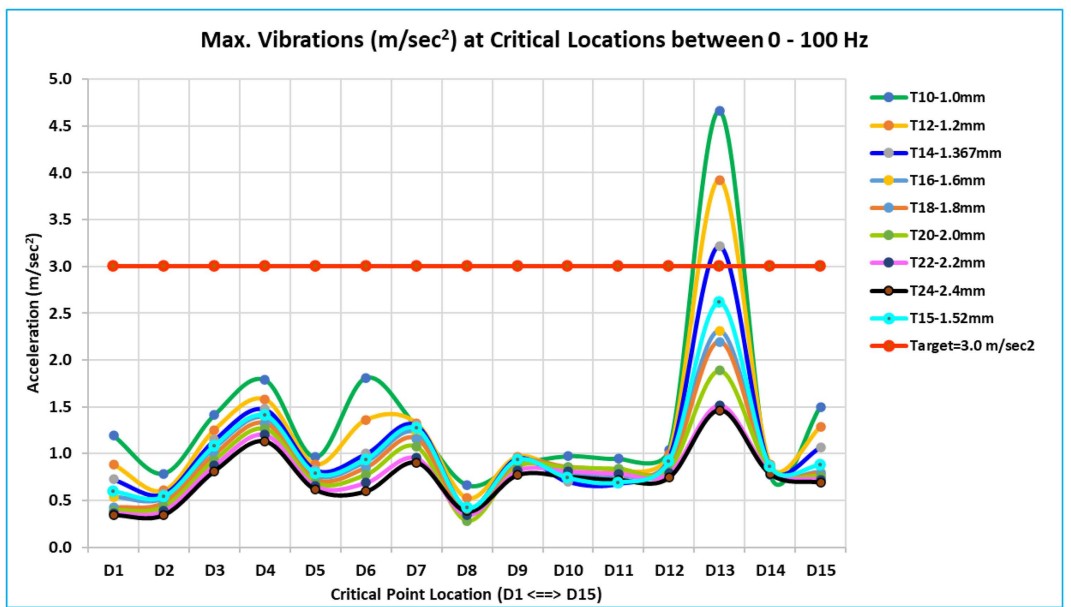

**Fig 17. Maximum Vibration at Critical Location between 0–100 Hz.**

**Table 6. Final Results Comparison Baseline and Optimized Model with Target.**

| Parameter | Target | Baseline Model T14 = 1.367 mm | Optimized Model T15 = 1.52 mm |
|---|---|---|---|
| 1st Global Lateral Bending Mode | > 26 Hz | 26.1 | 26.2 |
| 2nd Global Lateral Bending Mode | > 42 Hz | 42.5 | 43.2 |
| Min. Dynamic Stiffness between 20 Hz to 60 Hz | > 8.5 N/m | 7.5 | 9.3 |
| Max. Acceleration between 0 Hz to 100 Hz | < 3.0 m/sec$^2$ | 3.2 | 2.6 |

- The thickness of the inner panel has a linear relationship with the vibration, dynamic stiffness, and global bending mode frequencies of the door assembly.

- The required or optimum thickness of the door assembly can be determined using the linear trend line equation and the linear interpolation approach.

- The present model inner panel thickness of 1.367 mm has been changed to 1.52 mm in order to meet the required targets. All results are verified with the new thickness (Optimized) and results meets the target.

## Supporting information

**S1 Data. Minimal dataset.**
(DOCX)

## Acknowledgments

I wish to thank my guide, Dr. Kishor B. Waghulde, and Mr. R. Ramkumar for providing testing facilities at ARAI. I sincerely thank my research center, Dr. D. Y. Patil Institute of Technology, Pimpri, India, for providing the opportunity and facilitating

work on this highly sensitive and current topic. "Design and Development of Methodology to Simulate, Correlate, and Improve Car Door NVH Performance".

## Author contributions

**Conceptualization:** Pandurang Maruti Jadhav, Mohammad Israr.

**Data curation:** Pandurang Maruti Jadhav.

**Formal analysis:** Pandurang Maruti Jadhav, Kishor B. Waghulde.

**Investigation:** Venushree Khanke, Rupesh V. Bhortake, Prasad D. Kulkarni.

**Methodology:** Rupesh V. Bhortake, Prasad D. Kulkarni, Mohammad Israr.

**Resources:** Subhav Singh, Deekshant Varshaney.

**Software:** Md Irfanul Haque Siddiqui, Subhav Singh, Deekshant Varshaney.

**Supervision:** Md Irfanul Haque Siddiqui, Deekshant Varshaney, Mohammad Israr.

**Writing – original draft:** muasu jibrin musa.

**Writing – review & editing:** Mohammad Israr.

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
