## [Decision Letter · Decision Letter 0]

28 May 2025

Dear Dr. musa,

We look forward to receiving your revised manuscript.

Kind regards,

Khalil Abdelrazek Khalil, Ph.D.

Academic Editor

PLOS ONE

**Journal Requirements:**

1. When submitting your revision, we need you to address these additional requirements. Please ensure that your manuscript meets PLOS ONE's style requirements, including those for file naming. The PLOS ONE style templates can be found at https://journals.plos.org/plosone/s/file?id=wjVg/PLOSOne_formatting_sample_main_body.pdf and https://journals.plos.org/plosone/s/file?id=ba62/PLOSOne_formatting_sample_title_authors_affiliations.pdf 2. We suggest you thoroughly copyedit your manuscript for language usage, spelling, and grammar. If you do not know anyone who can help you do this, you may wish to consider employing a professional scientific editing service.  The American Journal Experts (AJE) (https://www.aje.com/) is one such service that has extensive experience helping authors meet PLOS guidelines and can provide language editing, translation, manuscript formatting, and figure formatting to ensure your manuscript meets our submission guidelines. Please note that having the manuscript copyedited by AJE or any other editing services does not guarantee selection for peer review or acceptance for publication.  Upon resubmission, please provide the following: The name of the colleague or the details of the professional service that edited your manuscript A copy of your manuscript showing your changes by either highlighting them or using track changes (uploaded as a *supporting information* file) A clean copy of the edited manuscript (uploaded as the new *manuscript* file) 3. Thank you for stating the following in your Competing Interests section:  “No” Please complete your Competing Interests on the online submission form to state any Competing Interests. If you have no competing interests, please state "The authors have declared that no competing interests exist.", as detailed online in our guide for authors at http://journals.plos.org/plosone/s/submit-now This information should be included in your cover letter; we will change the online submission form on your behalf. 4. We note that your Data Availability Statement is currently as follows: All relevant data are within the manuscript and its Supporting Information files. Please confirm at this time whether or not your submission contains all raw data required to replicate the results of your study. Authors must share the “minimal data set” for their submission. PLOS defines the minimal data set to consist of the data required to replicate all study findings reported in the article, as well as related metadata and methods (https://journals.plos.org/plosone/s/data-availability#loc-minimal-data-set-definition). For example, authors should submit the following data: - The values behind the means, standard deviations and other measures reported;- The values used to build graphs;- The points extracted from images for analysis. Authors do not need to submit their entire data set if only a portion of the data was used in the reported study. If your submission does not contain these data, please either upload them as Supporting Information files or deposit them to a stable, public repository and provide us with the relevant URLs, DOIs, or accession numbers. For a list of recommended repositories, please see https://journals.plos.org/plosone/s/recommended-repositories. If there are ethical or legal restrictions on sharing a de-identified data set, please explain them in detail (e.g., data contain potentially sensitive information, data are owned by a third-party organization, etc.) and who has imposed them (e.g., an ethics committee). Please also provide contact information for a data access committee, ethics committee, or other institutional body to which data requests may be sent. If data are owned by a third party, please indicate how others may request data access. 5. PLOS requires an ORCID iD for the corresponding author in Editorial Manager on papers submitted after December 6th, 2016. Please ensure that you have an ORCID iD and that it is validated in Editorial Manager. To do this, go to ‘Update my Information’ (in the upper left-hand corner of the main menu), and click on the Fetch/Validate link next to the ORCID field. This will take you to the ORCID site and allow you to create a new iD or authenticate a pre-existing iD in Editorial Manager.

Reviewers' comments:

Reviewer's Responses to Questions

**Comments to the Author**

1. Is the manuscript technically sound, and do the data support the conclusions?

Reviewer #1: Partly

Reviewer #2: No

2. Has the statistical analysis been performed appropriately and rigorously?

Reviewer #1: N/A

Reviewer #2: No

3. Have the authors made all data underlying the findings in their manuscript fully available?

Reviewer #1: No

Reviewer #2: No

4. Is the manuscript presented in an intelligible fashion and written in standard English?

Reviewer #1: No

Reviewer #2: No

**Reviewer #1:**  1. “…the system-damping effect is ignored to avoid numerical complexity,” A big simplification. Please justify this assumption.

2. Accelerometer Characteristics (Sensitivity, Mass and Dynamic Range). Please describe.

3. As the accelerometer will typically have an increase in sensitivity at the high frequency end due to its resonance, its output will not give a true representation of the vibration at the measuring point at these frequencies. Please describe.

4. The method of mounting the accelerometer to the measuring point is one of the most critical factors in obtaining accurate results from practical vibration measurements. Mounting the Accelerometer. Please describe.

5. There are two basic types of filter used for the frequency analysis vibration signals. The constant bandwidth type filter, where the filter is a constant absolute bandwidth, for example 3 Hz, 10 Hz etc. and the constant percentage bandwidth filter where the filter bandwidth is a constant percentage of the tuned centre frequency, for example 3%, 10% etc. Constant Bandwidth or Constant Percentage Bandwidth Frequency Analysis?

6. An important property of modes is that any forced or free dynamic response of a structure can be reduced to a discrete set of modes. The modal parameters are: • Modal frequency • Modal damping • Mode shape. Please designate.

7. Ideally a mobility measurement should simply involve exciting the structure with a measurable force, measuring the response, and then calculating the ratio between the force and response spectra. In practice however, we are faced with a number of problems: • Mechanical noise in the structure, including non-linear behaviour • Electrical noise in the instrumentation • Limited analysis resolution To minimize these problems we have to apply some statistical methods to decide how to estimate an FRF from our measurements. Estimation from data containing random noise generally involves some form of averaging. What techniques can we use for averaging the output/input ratio?

8. Please compare your research results with those of other scientists.

**Reviewer #2: ** The paper presents a study on optimising the thickness of a car's inner door panel for enhanced stiffness and vibration control. Authors present an interesting work, nevertheless, consider addressing the following points to improve it [the meaning of (R) and (I) is defined at the end of the comments]:

1. (I) Document. Authors must take care with written language so they can convey what they really want. The importance of content deserves more careful and technical language.

2. (R) Abstract. The abstract should be more succinct.

3. (R) Keywords. Authors should avoid using words used in the title for broader content search.

4. (I) Page 2, Introduction, first sentence. Before the COVID-19 pandemic, were cars not commonly used? (reference to 1.)

5. (I) Page 2, Introduction, third sentence. “The car door is used to get in and out of the car while it is in use”. What do the authors mean? Getting in and out while the car is moving? (reference to 1.)

6. (I) … (reference to 1.)

7. (R) Authors should improve the state of the art.

8. (R) Symbols should be italicised.

9. (I) In equations, all symbols must be defined.

10. (R) In figures with more than one image, they must have their own identifiers [(a), (b),...] and be properly identified/described in figure’s caption.

11. (R) Table 2. Authors should better define the units in the second column.

12. (R) Figures 8, 9, 10, 11, 13 and 16. Y-axis values must be the same on graphs, for better comparison.

13. (R) All figures must be named in the text before they appear.

The developed work deserves a more careful presentation in this paper.

(S) Suggestion. The author can change it or not. Does not compromise scientific rigour or understanding of the document.

(R) Recommendation. It does not compromise scientific rigour, but it compromises the rigour of communication and understanding.

(I) Imposition. It compromises scientific rigour.

**Do you want your identity to be public for this peer review?** For information about this choice, including consent withdrawal, please see our Privacy Policy

Reviewer #1: **Yes: ** Krzysztof Szwajka

Reviewer #2: No

---

## [Author Response · Author response to Decision Letter 1]

12 Jun 2025

We have done all corrections as suggested

---

## [Decision Letter · Decision Letter 1]

29 Jun 2025

Dear Dr. musa,

Thank you for submitting your manuscript to PLOS ONE. After careful consideration, we feel that it has merit but does not fully meet PLOS ONE’s publication criteria as it currently stands. Therefore, we invite you to submit a revised version of the manuscript that addresses the points raised during the review process.

We look forward to receiving your revised manuscript.

Kind regards,

Khalil Abdelrazek Khalil, Ph.D.

Academic Editor

PLOS ONE

Journal Requirements:

Reviewers' comments:

Reviewer's Responses to Questions

**Comments to the Author**

Reviewer #1: All comments have been addressed

Reviewer #2: (No Response)

2. Is the manuscript technically sound, and do the data support the conclusions?

Reviewer #1: Yes

Reviewer #2: Yes

3. Has the statistical analysis been performed appropriately and rigorously?

Reviewer #1: Yes

Reviewer #2: Yes

4. Have the authors made all data underlying the findings in their manuscript fully available?

Reviewer #1: Yes

Reviewer #2: Yes

5. Is the manuscript presented in an intelligible fashion and written in standard English?

Reviewer #1: Yes

Reviewer #2: No

Reviewer #1: The authors have made the proposed corrections to the article. Thank you. I propose publishing the article.

Reviewer #2: The paper presents a study on optimising the thickness of a car's inner door panel for enhanced stiffness and vibration control. Authors present an interesting work, nevertheless, consider addressing the following points to improve it [the meaning of (R) and (I) is defined at the end of the comments]:

1. (R) Document. Authors must take care with written language. The importance of content deserves more careful and technical language. There was a noticeable improvement in writing. The document is understandable despite being written in informal language.

2. (R) Document. Change “...between... to…” to “...between... and...”.

3. (S) Keywords. Authors should avoid using words used in the title for broader content search. For example (it is not mandatory to use these examples): “dynamic analysis”, “FEM”, “modal frequency”,...

4. (S) Line 68. There is a disagreement in the requirement to separate the symbol "%" from the respective value. In my opinion, it should be separated.

5. (R) Lines 70, 76, 99, 102, 112, 117, 118, 120, 123, 125, 167. The sentence needs to be improved. Authors should begin the sentence with "AUTHOR studied/analysed/..."

6. (R) Line 108. Change “it’s” to “its”.

7. (R) Line 163. Remove the worth “the”.

8. (R) Figures 2, 6, 8-11, 13-16. In figures with more than one image, they must have their own identifiers [(a), (b),...] and be properly identified/described in figure’s caption and text.

9. (I) Line 187. The sentence needs to be improved.

10. (R) Line 191. Change “1.00” to “1.0”.

11. (R) Line 192. Change “steps of 0.2 mm (1.0, 1.2, 1.4, 1.6, 1.8, 2.0, 2.2, and 2.4 mm)” to “ steps of 0.2 mm, (1.0, 1.2, 1.4, 1.6, 1.8, 2.0, 2.2 and 2.4) mm”.

12. (I) Line 237. “… were taken in all three XY directions…”. Three (XYZ) or two (XY) directions? Which direction Y is?

13. (R) Table 2. SI density units: [kg/m^3].

14. (R) Table 3. Lateral, vertical and torsional bending units?

15. (R) Figures 8 to 12. SI stiffness units: N/m. You can solve by changing the axis title to “Dynamic stiffness (x 10^6 N/m)”.

16. (R) All figures must be named in the text before they appear.

17. (R) Lines 351 and 352. change “(T10, to T24)” to “(T10 to T24)” and “(D1, to D15)” to “(D1 to D15)”.

18. (S) Line 385. Improving English writing. For example, “The following are the interpretations after an…”.

19. (R) Line 489. Improving English writing. “… in Tables 3, 4, and 5.” or “… in Tables 3 to 5.”

20. (I) In finite element analysis, what is the dimension of the finite elements? For a good analysis, there should be at least 7 finite elements at one wavelength (there is bibliography on this subject). Has this been verified or has any sensitivity analysis been done on the finite element dimension?

(S) Suggestion. The author can change it or not. Does not compromise scientific rigour or understanding of the document.

(R) Recommendation. It does not compromise scientific rigour, but it compromises the rigour of communication and understanding.

(I) Imposition. It compromises scientific rigour.

**Do you want your identity to be public for this peer review?** For information about this choice, including consent withdrawal, please see our Privacy Policy

Reviewer #1: **Yes: ** Szwajka Krzysztof

Reviewer #2: No

---

## [Decision Letter · Decision Letter 2]

24 Jul 2025

Dear Dr. musa,

Thank you for submitting your manuscript to PLOS ONE. After careful consideration, we feel that it has merit but does not fully meet PLOS ONE’s publication criteria as it currently stands. Therefore, we invite you to submit a revised version of the manuscript that addresses the points raised during the review process.

We look forward to receiving your revised manuscript.

Kind regards,

Khalil Abdelrazek Khalil, Ph.D.

Academic Editor

PLOS ONE

Journal Requirements:

1.If the reviewer comments include a recommendation to cite specific previously published works, please review and evaluate these publications to determine whether they are relevant and should be cited. There is no requirement to cite these works unless the editor has indicated otherwise. 

Reviewers' comments:

Reviewer's Responses to Questions

**Comments to the Author**

Reviewer #2: (No Response)

2. Is the manuscript technically sound, and do the data support the conclusions?

Reviewer #2: Yes

3. Has the statistical analysis been performed appropriately and rigorously?

Reviewer #2: Yes

4. Have the authors made all data underlying the findings in their manuscript fully available?

Reviewer #2: Yes

5. Is the manuscript presented in an intelligible fashion and written in standard English?

Reviewer #2: No

Reviewer #2: The paper presents a study on optimising the thickness of a car's inner door panel for enhanced stiffness and vibration control. Authors present an interesting work, nevertheless, consider addressing the following points to improve it.

(Recommendation for language clarity) Page 3, paragraph 4; page 4, last sentence of paragraph 1, 1st sentence of paragraph 2, 3th sentence of paragraph 3; Page 5, last sentence of paragraph 1. The sentences need to be improved. Authors should begin the sentences with "AUTHOR studied/analysed/…".

Authors should be congratulated for their effort and remarkable progress in written language from the first version to this one.

**Do you want your identity to be public for this peer review?** For information about this choice, including consent withdrawal, please see our Privacy Policy

Reviewer #2: No

---

## [Decision Letter · Decision Letter 3]

14 Aug 2025

Optimization of Inner Panel Thickness for Enhanced Stiffness and Vibration Control in Car Door Assemblies

PONE-D-25-18089R3

Dear Dr. musa,

We’re pleased to inform you that your manuscript has been judged scientifically suitable for publication and will be formally accepted for publication once it meets all outstanding technical requirements.

Kind regards,

Khalil Abdelrazek Khalil, Ph.D.

Academic Editor

PLOS ONE

Additional Editor Comments (optional):

Reviewers' comments:

Reviewer's Responses to Questions

**Comments to the Author**

Reviewer #2: All comments have been addressed

2. Is the manuscript technically sound, and do the data support the conclusions?

Reviewer #2: Yes

3. Has the statistical analysis been performed appropriately and rigorously?

Reviewer #2: Yes

4. Have the authors made all data underlying the findings in their manuscript fully available?

Reviewer #2: Yes

5. Is the manuscript presented in an intelligible fashion and written in standard English?

Reviewer #2: Yes

Reviewer #2: The paper presents a study on optimising the thickness of a car's inner door panel for enhanced stiffness and vibration control. The work is interesting and the paper well written (the written English could be improved but the document is intelligible). Good work.

**Do you want your identity to be public for this peer review?** For information about this choice, including consent withdrawal, please see our Privacy Policy

Reviewer #2: No

---

## [Editor Report · Acceptance letter]

PONE-D-25-18089R3

PLOS ONE

Dear Dr. musa,

I'm pleased to inform you that your manuscript has been deemed suitable for publication in PLOS ONE. Congratulations! Your manuscript is now being handed over to our production team.

Kind regards,

on behalf of

Dr. Khalil Abdelrazek Khalil

Academic Editor

PLOS ONE